# *Helicobacter trogontum* Bacteremia and Lower Limb Skin Lesion in a Patient with X-Linked Agammaglobulinemia—A Case Report and Review of the Literature

**DOI:** 10.3390/pathogens11111247

**Published:** 2022-10-27

**Authors:** Lasse Fjordside, Caroline Herløv, Camilla Heldbjerg Drabe, Leif Percival Andersen, Terese L. Katzenstein

**Affiliations:** 1Department of Infectious Diseases, University Hospital, Rigshospitalet, 2100 Copenhagen, Denmark; 2Department of Clinical Microbiology, University Hospital, Rigshospitalet, 2100 Copenhagen, Denmark

**Keywords:** *Helicobacter trogontum*, *Flexispira rappini*, non-*Helicobacter-pylori*-helicobacters, enterohepatic helicobacters, X-linked agammaglobulinemia, cellulitis, bacteremia, zoonotic bacterial infection, opportunistic infections

## Abstract

We describe the first case of infection with *Helicobacter trogontum* in a patient with X-linked agammaglobulinemia. A 22-year-old male with X-linked agammaglobulinemia presented with fever, malaise and a painful skin lesion on the lower left extremity. Spiral shaped Gram-negative rods were isolated from blood cultures and later identified as *Helicobacter trogontum*. The patient was treated with various intravenous and oral antibiotic regimens over a period of 10 months, each causing seemingly full clinical and paraclinical remission, yet several episodes of relapse occurred after cessation of antibiotic treatment. The review of the literature showed that only a few cases of infections with enterohepatic helicobacters belonging to the *Flexispira rappini* taxons have previously been reported. The majority of cases included patients with X-linked agammaglobulinemia and the symptomatology and course of disease were similar to the case described here. Infections with enterohepatic helicobacters, including *Helicobacter trogontum*, should be considered in patients with X-linked agammaglobulinemia presenting with fever, malaise and skin lesions. Careful cultivation and microbiological investigation are essential to determine the diagnosis and a long treatment period of over 6 months must be expected for successful eradication.

## 1. Introduction

### 1.1. Immunology

X-linked agammaglobulinemia (XLA) is a humoral immunodeficiency caused by a loss-of-function mutation in a gene located on the long leg of the X-chromosome (Xq21.3-q22) [1,2]. This gene is responsible for the production of an enzyme known as Bruton’s tyrosine kinase (Btk), which is paramount for the maturation of B-cells [1]. Patients with XLA are characterized by a depletion of all subgroups of immunoglobulins making them prone to a series of infections, especially respiratory [3], causing considerable morbidity and mortality [4]. Treatment with immunoglobulin replacement therapy is used to reduce the infectious burden and lower the risk of chronic lung injury and has substantially improved the long-term prognosis [1,5]. The causative organisms are most often encapsulated bacteria such as *Haemophilus influenzae*, *Streptococcus pneumoniae* and *Staphylococcus aureus*. However, enterohepatic helicobacter species have more recently been reported as an important source of opportunistic infections in patients with XLA.

### 1.2. Microbiology

The *Helicobacter* genus currently entails 35 named species and several candidate taxons and can be divided into two groups according to their biological niches: the gastric helicobacters and the enterohepatic helicobacters. Among the gastric helicobacters, *Helicobacter pylori* has claimed its fame for the revolutionizing discovery of its role as etiological agent to gastric ulcers and gastric cancers [6]. Many of the enterohepatic helicobacters have animal intestines as their natural habitats but may cause infections in humans [7]. “*Flexispira rappini*” is a provisional name for Gram-negative microaerophilic, motile, fusiform-shaped organisms with spiral periplasmic fibers and bipolar tufts of sheathed flagella [8]. The phylogenetic position of “*Flexispira rappini*” falls within the genus *Helicobacter* according to DNA-rRNA hybridization [9] and 16S rRNA sequencing [10,11]. It consists of several *Helicobacter* taxa, named *Flexispira* taxon 1 to *Flexispira* taxon 10 [11]. Many of the taxons have no valid name, but taxon 6 has been named *Helicobacter trogontum* and taxon 9 *Helicobacter bilis*. The remaining 8 taxons are still called by numbers [11] (Figure 1).

This is the first report of infection with *Helicobacter trogontum* in a patient with X-linked agammaglobulinemia and, to our knowledge, only the second report of human infection with this pathogen [12].

### 1.3. Case Presentation

The patient is a 22-year-old male of Latin-American heritage, who was diagnosed with XLA at age 8 due to repeated severe pneumonias.

Two of the patient’s brothers had died from infectious diseases before age 10 as did two of his mother’s brothers. The patient has one living brother, who does not suffer from XLA.

At age 15 the patient moved with his family to the United States of America to gain access to immunoglobulin treatment, not accessible in the home country at the time. In the United States he was treated with intravenous immunoglobulins every three months.

After a few years, the patient moved to Europe and settled in Denmark in 2021, where he found work on a pig farm. The work included feeding and mucking out. The patient reported using disposable full-body suits, gloves and thorough hand hygiene after work.

Due to XLA he was referred to our out-patient-clinic at the Department for Infectious Diseases at Copenhagen University Hospital, Rigshospitalet where he made his first visit in June 2021. The diagnosis was genetically verified through whole genome sequencing showing a pathologic nonsense hemizygotic variant in the Btk gene (p.(Trp251Ter)). Gammaglobulin treatment was re-initiated. We were not able to clarify exactly how long the patient had been without immunoglobulin treatment, but our best estimate is around six months. The delay was assumed to be caused by the time it took to settle in, get a general practitioner appointed and be referred to our department.

In October 2021 the patient presented at the outpatient clinic with localized erythema on the tibial region of his left leg and was treated with oral penicillin for presumed erysipelas and experienced full remission. Yet, a couple of weeks later the erythema returned and the patient was prescribed a course of penicillin and dicloxacillin, again causing full remission. In December 2021, however, the patient presented to the emergency room of a peripheral hospital with fever, malaise and a painful red skin lesion on his lower left extremity and was subsequently transferred to our department for further assessment.

The main finding of the initial physical examination was the skin lesion described as a red-to-purple, elevated, exudative lesion, measuring 5 × 7 cm, located on the tibial region of the left lower extremity. Vitals showed a heart rate of 90 bpm, a blood-pressure of 110/53 mmHg, a respiratory frequency of 18/min, oxygen saturation of 98% and a temperature of 39.3 °C (102.7 F). Bloodwork showed mild anemia with a hemoglobin concentration of 7.4 mmol/L (ref.: 8.3–10.5), raised inflammatory marks with a C-reactive-protein of 132 mg/L (ref.: <10) (Figure 2A), a slightly elevated neutrophile count of 6.04 × 10^9^/L (ref.: 1.6–5.9) and a mild thrombocytosis of 419 × 10^9^/L (ref.: 145–390). Immunoglobulin levels were low with IgG of 5.3 g/L (ref.:6.1–14.9) (Figure 2A), IgM and IgA both <0.05 g/L. An X-ray of the leg showed no signs of osteomyelitis. A skin biopsy of the lesion showed unspecific inflammation. Spiral shaped Gram-negative rods were detected in 1 of 2 aerobic blood cultures.

The patient was treated with intravenous piperacillin-tazocin 4 g × 4 daily and oral doxycycline 500 mg twice daily for 10 days and discharged in full clinical and paraclinical remission without supplemental antibiotic treatment (Figure 2B). Two weeks later a nested 16s-RNA-PCR identified the Gram-negative rods as *Helicobacter trogontum*. In vitro susceptibility testing of the bacterial strain isolated from the patient’s blood showed no resistance to any of the antibiotics tested (Figure 3).

In the 6 months following the initial admission for the infection, two episodes of relapse with increased C-reactive-protein and return of the red, swollen and painful skin lesion, occurred. Both were treated with 3–4 weeks courses of amoxicillin and ciprofloxacin. The first treatment led to full remission of the wound and normalized C-reactive-protein but during the second treatment, the lesion worsened, and C-reactive-protein increased. The treatment was changed to doxycycline, but due to continuous worsening the patient was re-admitted and the treatment regimen was changed to moxifloxacin 400 mg once daily and rifampicin 600 mg twice daily (Figure 2B). During admission a dermatological evaluation deemed the wound highly suggestive of pyoderma gangraenosum and local steroid treatment was prescribed (Figure 4). Infection counts normalized, the skin lesion got significantly better, and the patient was discharged with continued treatment with moxifloxacin, rifampicin, and topical steroid.

## 2. Review of the Literature

Different databases were used including PubMed, PMC and Embase. At first 10 different search strings were made, one for each taxon of *Flexispira rappini* combined with human infection. No hits were found except from the search with *Flexispira rappini* and human infection. Therefore, one search string was used “*Flexispira rappini* and human infection”.

One hundred and sixty-one hits were found in PubMed, nine of them were relevant, 102 hits were found in PMC of which six were relevant (all also identified in the PubMed search). No hits were found in Embase. The variation (Table 1) illustrates the possible advantage of including several databases in the search strategy.

From the results in Table 2, it can be summarized that all the identified studies have investigated cases related to an association between human infection and *Flexispira rappini* infection.

*Flexispira rappini* appears in all studies but in some cases in combination with other bacteria. It appears that most of the patients described were immuno-incompetent and 5/9 had XLA.

## 3. Discussion

This case report describes the first documented infection with *Helicobacter trogontum* in a patient with XLA and only the second ever reported human infection with this fastidious and problematic pathogen. The case underlines the significant challenges with diagnosing and treating infections with enterohepatic helicobacter species.

There is only a limited number of case reports on human infection with enterohepatic helicobacters belonging to one of the 10 *Flexispira rappini* taxons. All previously documented human infections with this group of helicobacters have been caused by *Helicobacter bilis* (*Flexispira rappini* taxon 8). Interestingly, the majority of case reports involved patients with XLA and with similar clinical features, including the characteristic skin lesions also documented in this case report.

Other enterohepatic helicobacter species than those included in the *Flexispira rappini* taxons may cause infections with similar clinical features in both immunocompetent and immune-incompetent cases, including patients with XLA. Of these the most frequently reported is *Helicobacter cinaedi* [22,23].

Patients with XLA seem to be especially susceptible to infections with enterohepatic helicobacter species including *Helicobacter trogontum*, *Helicobacter bilis* and *Helicobacter cinaedi* [22]. These infections often cause skin lesions that mimic erythema nodosum or pyoderma gangrenosum and are most often located on the lower extremities [22].

When patients with XLA present with cellulitis or erythema, infection with enterohepatic helicobacter species should be considered. If suspected, blood cultures should be sent for careful helicobacter-specific cultivation, 16s-RNA-PCR and if available MALDI-TOF analysis. A skin biopsy should be obtained and examined for the presence of spiral shaped rods (e.g., using the Warthin–Starry silver stain [22]) and sent for 16s-RNA-PCR, and a PCR for enterohepatic helicobacters should be conducted on a fecal swab. The treatment should consist of antibiotics from at least two different classes to reduce the risk of developing resistance. The strain under investigation here showed full susceptibility to a wide range of antibiotics (Figure 4) and with an expected course of treatment in the range of 6 months [24], factors like adherence and tolerability should be considered when deciding on a drug combination. From our case it can be deducted that intravenous treatment is not mandatory in order to achieve apparent remission, yet whether it is superior in causing full eradication is uncertain. Supplemental topical steroid treatment may be used if pyoderma gangraenosum is suspected, but lesions seem to respond well to antibiotic treatment alone too.

Infections with enterohepatic helicobacter species are truly fascinating and raise many intriguing questions of which only a few will be discussed here.

### 3.1. Why Do Patients with XLA Seem to Be Especially Susceptible to Infection with Enterohepatic Helicobacters?

Under normal physiological conditions, large amounts of IgA are secreted into the intestinal lumen from plasma B-cells residing in the germinal centers of the so-called Peyer’s patches of the intestinal wall. The role of IgA in the intestinal lumen is to prevent intestinal bacteria and their related toxins from getting in contact with the luminal epithelium and translocating to the mesenteric compartment [25]. The lack of intestinal IgA in patients with XLA may thus contribute to the susceptibility to disseminated infection. This is in line with the fact that adequate IVIG treatment and normal serum-levels of IgG do not seem to fully protect against enterohepatic helicobacter infections in patients with XLA [22]. The insufficient protection from intravenous immunoglobulin therapy, could indicate a lack of pathogen specific IgG in the donor-pool. However, lack of IgA is unlikely to be the only mechanism involved. The absence of IgM could also potentially contribute to the lack of containment of the pathogen [26] but its significance is uncertain. Apart from B-cell maturation, Btk is also involved in a vast amount of other processes and signaling pathways in innate immunity, including pathogen recognition via Toll-Like-Receptors (TLRs) [27]. Failure in pathogen recognition could protect enterohepatic helicobacters from being killed and confuse the downstream immunological response allowing translocation to the mesenteric compartment and subsequent continuous proliferation ultimately resulting in bacteremia. Another potential contribution to the risk of translocation could be entailed in the virulence factors of the bacteria. *Helicobacter pylori* is able to open epithelial tight junctions in gastric mucosa via a urease mediated internalization of occludin [28]. It is uncertain whether some enterohepatic helicobacter species are able to do the same, but they share the necessary enzymes [29]. The bacterial translocation from the intestinal lumen to the mesenteric lymphoid system could thus be obtained through piggybacking dendritic cells or alternatively through direct bacterial disruption of the epithelial barrier function. It has previously been suggested that systemic enterohepatic helicobacter infections begin as gut infections and disseminate from there [30]. In addition, since most enterohepatic helicobacter species have animal intestines as their natural habitat [7], we hypothesize that infections with enterohepatic helicobacters begin with the incidental oral ingestion of animal-gut derived enterohepatic helicobacter species which in some individuals may cause a local colonization and enteritis and in other susceptible individuals go on to cause disseminated disease through bacterial translocation from the gut to the lymphatic and vascular system.

### 3.2. What Is the Nature of the Skin Lesions?

The characteristic skin lesions seem to be the hallmark clinical finding in patients with disseminated enterohepatic helicobacter infections. The consistency in location, size, development and appearance of the lesions across the different case reports is noticeable and suggests a common pathogenesis. Interestingly, the skin lesions associated with enterohepatic helicobacter infection share several characteristics with erythema nodosum and pyoderma gangraenosum including: clinical appearance, location and the common association with gastrointestinal inflammation [22,31,32,33,34]. We hypothesize that these conditions have a shared pathogenesis and that enterohepatic helicobacters could be the causative agents of some of the otherwise considered idiopathic cases of erythema nodosum and pyoderma gangraenosum. Spiral shaped rods have been identified through staining of skin biopsies from active skin lesions in some patients with enterohepatic helicobacter infections [21], though most case reports fail to identify any microorganisms in skin biopsies [16,23,30,31]. It can therefore not be completely ruled out that the lesions could indeed be infectious. However, a reactive inflammatory etiology caused by immune modulation related to the enterohepatic helicobacter infection similar to the development of erythema nodosum in aftermath of an enteric infection (e.g., the closely related campylobacter species) does seem more likely. We have not found any plausible explanation for the almost exclusive restriction of the skin lesions to the extensor side of the lower extremities.

### 3.3. Could Enterohepatic Helicobacters Be the Causative Agents of Other Diseases?

Because helicobacters are so difficult to isolate and cultivate from regular clinical specimens, infections with enterohepatic helicobacters are likely under-diagnosed. The significant similarities of the skin lesions associated with enterohepatic helicobacter infections and erythema nodosum and pyoderma gangrenosum raises the question whether enterohepatic helicobacters could be the causative agents of these idiopathic conditions [32]. Further, since both erythema nodosum and pyoderma gangrenosum are associated with inflammatory bowel disease (IBD) [31], and enterohepatic helicobacters can cause enteritis [30] it has also been hypothesized that enterohepatic helicobacters could play a role in the development of IBD [35,36]. Patients with XLA have a significantly increased risk of IBD [37], yet none of the patients identified in this review showed symptoms of enteritis. *Helicobacter pylori* belongs to the gastric helicobacters and is associated with the development of gastric ulcers and gastric cancers [6]. Patients with primary immune deficiency are known to carry a significantly increased risk of cancers, including a 50 times greater risk of developing a gastric adenocarcinoma compared to immunocompetent individuals [38]. In this light, it could be interesting to investigate the role of enterohepatic helicobacters in the development of hepatic and colorectal cancers.

## Figures and Tables

**Figure 1 pathogens-11-01247-f001:**
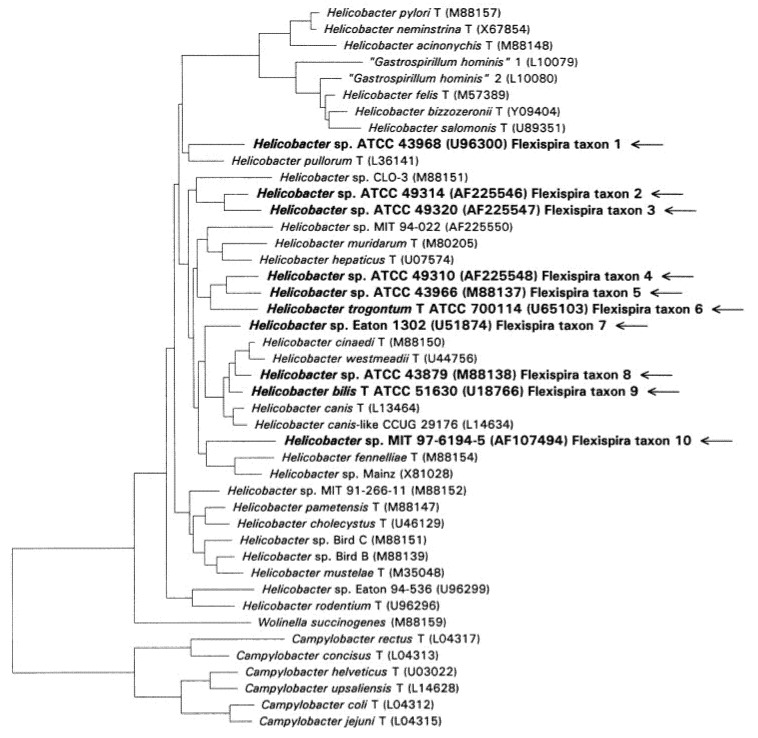
Phylogenetic tree of *Helicobacters* and *Campylobacters*. Phylogenetic tree of *Helicobacters* and *Campylobacters*. The 10 *Flexispira rappini* taxons are highlighted and marked with arrows [11].

**Figure 2 pathogens-11-01247-f002:**
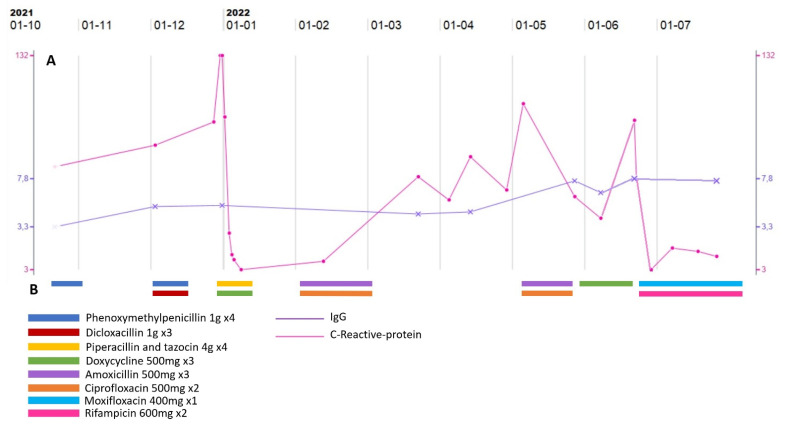
Timeline of the course of disease. (**A**) Shows the development of C-reactive-protein and IgG over the course of the disease. Time is on the X-axis and is depicted in the top. The grey vertical lines mark each month. The fluctuations in C-reactive-protein reflect the episodes of clinical relapses. The level of IgG is held relatively stable within the therapeutic range throughout the entire period and does not seem to affect the course of the disease. (**B**) Depicts the different treatment-regiments. Color codes are translated in the legends.

**Figure 3 pathogens-11-01247-f003:**
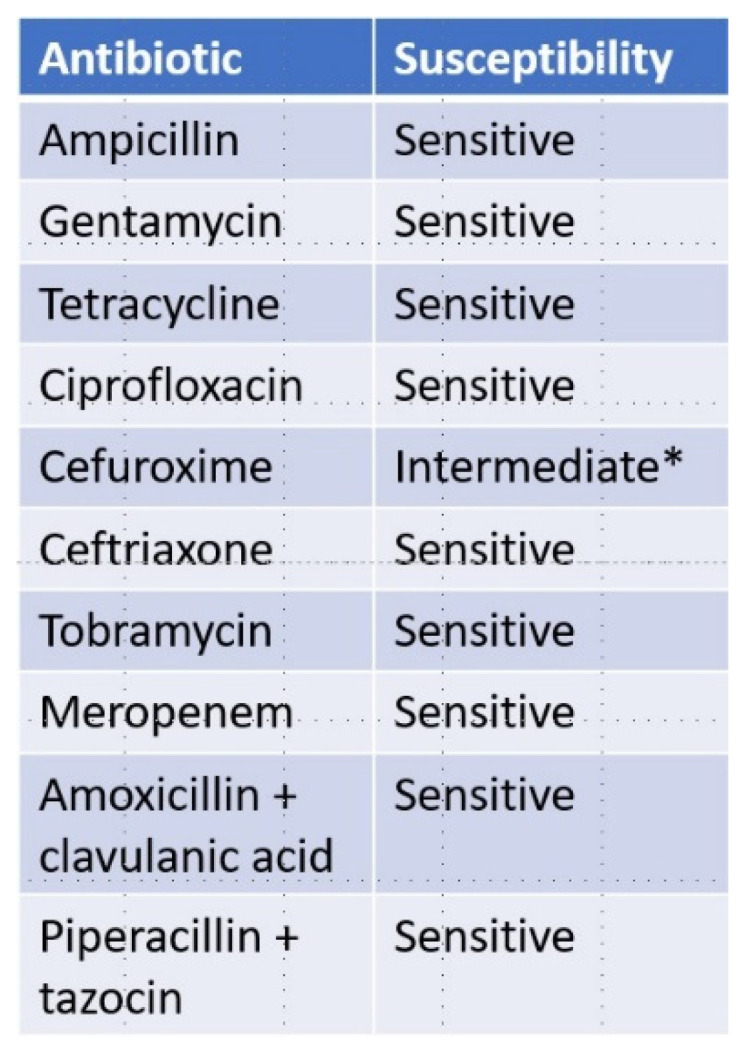
The susceptibility of the isolated strain of *helicobacter trogontum* to the tested antibiotics. * Susceptible at increased dosage.

**Figure 4 pathogens-11-01247-f004:**
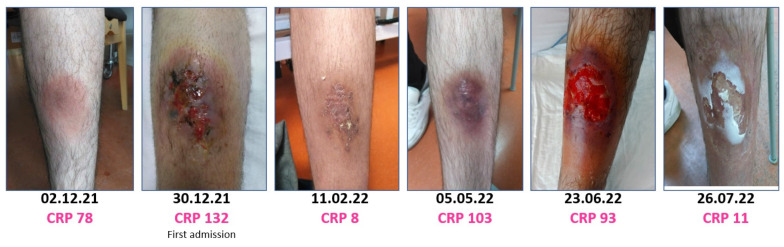
A series of selected clinical photos showing the fluctuation of the skin lesion over time and the corresponding C-reactive-protein values.

**Table 1 pathogens-11-01247-t001:** Search strategy.

Database	Search String	Hits	Relevant Hits
PubMed	*Flexispira rappini* and human infection	161	9
PMC	*Flexispira rappini* and human infection	102	6 (these were also found in PubMed)
Embase	*Flexispira rappini* and human infection	0	0

**Table 2 pathogens-11-01247-t002:** Results from literature search of human infection and *Flexispira rappini*.

Year	Author	Material	Found	Disease of the Patient	Disease as Cause of Infection
1998	Tee et al. [13]	Blood culture	*Flexispira rappini*	Appendectomy and recurrent chest infections over the last 2 years	Pneumonia/Bacteremia
1999	Sorlin et al. [14]	Blood culture	*Flexispira rappini*	End-stage renal failure, chronic pancreatitis and secondary diabetes mellitus	Bacteremia
1999	Weir et al. [15]	Blood culture	*Flexispira rappini* (possible new taxon)	X-linked agammaglobulinemia	Persistent sepsis with leg swelling
2000	Cuccherini et al. [16]	Blood/skin	*Flexispira rappini*	X-linked agammaglobulinemia	Bacteremia, skin/bone infection
2000	Han et al. [17]	Pus from the abscess	*Flexispira rappini, Helicobacter bilis, and Helicobacter* sp. *strain Mainz*	X-linked agammaglobulinemia	Abdominal abscess
2001	Iten et al. [18]	Blood culture	*Flexispira rappini* taxon 8	Healthy but skin lesion in relation to travelling	Sweats, chills, fever, diffuse arthralgias, and painful legs
2001	Gerrard et al. [19]	Blood samples	*Flexispira rappini and Helicobacter canis*	X-linked agammaglobulinemia	Recurrent bacteremia and multifocal lower limb cellulitis
2005	Brachet-Castang et al. [20]	Blood culture	*Flexispira rappini* taxon 8	Common Variable Immunodeficiency	Bacteremia
2010	R. Murray et al. [21]	Blood culture	*Flexispira rappini* Taxon 8	X-linked agammaglobulinemia	Chronic pyoderma gangrenosum-like leg ulcer

## Data Availability

Not applicable.

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
