# Peer review of "Helicobacter trogontum Bacteremia and Lower Limb Skin Lesion in a Patient with X-Linked Agammaglobulinemia—A Case Report and Review of the Literature"

_pathogens, 2022, doi:10.3390/pathogens11111247_

Round 1
Reviewer 1 Report
This manuscript details a case and some literature review on an enigmatic and highly problematic pathogen. Overall the case is well presented and the figures and legends amplify the message. I have a few minor criticisms and suggestions.
1) The taxonomy of Helicobacter confuses everyone. It would be useful to have a figure of the taxonomy.
2) Please include BTK mutation.
3) Typos in case presentation: "blood-works showed mild anemia". Should be Blood work. Just below it: "Raised infection marks with a C'reactive protein...". Should be raised inflammatory marks with a C-reactive protein...
4) The in vitro resistance is given as part of the discussion- I think it belongs more naturally in the case discussion.
5) The legend for the treatments in Figure 1 is not given.
6) I do not the search methods warrants a table. A one sentence description is probably enough.
7) Table 2 should include the treatments and relapses to be maximal useful. It would also be helpful to ahve a summary (%XLA, % limb % organ) and I think you should consider including Helicobacter cinaedi.
8) Helicobacter trogontum has never been seen in human disease?
9) The issue of the pyoderma diagnosis requires more thought and discussion. It was felt to be an aberrant healing process? A reactive inflammatory process? Just a pathological representation of the underling infection?
10) The discussion on why this occurs in XLA is limited to a discussion of IgA and compromised barrier function. Lack of IgM and possibly the role of BTK in myeloid TLR responses are at least theoretical considerations. There is some effort to link it to IBD but no mention of the high rate of IBD in XLA. Furthermore- as far as I am aware, none of the previous XLA cases had Helicobacter and IBD.
Author Response
1) The taxonomy of Helicobacter confuses everyone. It would be useful to have a figure of the taxonomy.
Agreed! It has now been included.
2) Please include BTK mutation.
Has now been included.
3) Typos in case presentation: "blood-works showed mild anemia". Should be Blood work. Just below it: "Raised infection marks with a C'reactive protein...". Should be raised inflammatory marks with a C-reactive protein...
Has now been corrected.
4) The in vitro resistance is given as part of the discussion- I think it belongs more naturally in the case discussion.
A valid point. It has been relocated.
5) The legend for the treatments in Figure 1 is not given.
Mistake. It has now been added.
6) I do not the search methods warrants a table. A one sentence description is probably enough.
We have chosen to maintain the table, but agree that its necesity is questionable.
7) Table 2 should include the treatments and relapses to be maximal useful. It would also be helpful to ahve a summary (%XLA, % limb % organ) and I think you should consider including Helicobacter cinaedi.
We chose to focus on flexispira rappini in the review of the literature as we felt that, from a microbiological point of view, the inclusion of helicobacter cinaedi could have blurred the interpretation. We agree with the clinical relevance and hope that other authors will be able to gather all relevant articles in a comprehensive review similar by to the recent efforts from Romo-Gonzales et al.
8) Helicobacter trogontum has never been seen in human disease?
To our knowledge, only one other human case has been reported . The mentioning of it is relevant and i have now been added, including its reference.
9) The issue of the pyoderma diagnosis requires more thought and discussion. It was felt to be an aberrant healing process? A reactive inflammatory process? Just a pathological representation of the underling infection?
The similarity of the skin lesions in appearence, size and location across the case reports (including those with helicobacter cineadi) is astounishing, but it is challenging to propose a coherent physiological explanation to their pathogenesis. We have tried to ellaborate a bit more on this subject in our revision.
10) The discussion on why this occurs in XLA is limited to a discussion of IgA and compromised barrier function. Lack of IgM and possibly the role of BTK in myeloid TLR responses are at least theoretical considerations. There is some effort to link it to IBD but no mention of the high rate of IBD in XLA. Furthermore- as far as I am aware, none of the previous XLA cases had Helicobacter and IBD.
We agree that there is undoubtedly many different factors contributing to the pathogenesis of disseminated enterohelicobacter infections. We have elaborated this part of the discussion a bit more.
Reviewer 2 Report
This article describes for the first time a Helicobacter trogontum infection in a patient with X-linked agammaglobulinemia.
This clinical case is very well written, the scientific significance is substantial, making a singular contribution to the sometimes difficult diagnoses of opportunistic infections during this disorder.
There are a certain number of editorial comments to be made.
Line 66, the authors report treatment with intravenous immunoglobulin every 3 months instead of every 3 weeks.
Some English corrections are required, for example on lines 68 and 74.
Please note a small typographical error on line 160.
With regard to the clinical description, we would like to know the nature and duration of the interruption of immunoglobulin replacement therapy between the patient's leaving the United States and arriving in Denmark, as well as the chronology of the interruption prior to the occurrence of the infection.
It would be helpful to clarify the nature of his work on the pig farm because pigs are known to be carriers of certain strains of Helicobacter.
About Figure 1, the CRP and IgG assay curves should be more specifically captioned. In this figure or in the text, the duration of antibiotic treatment should be more clearly indicated.
In the discussion, it seems important to broaden the reflections on IgA deficiency. It is true that agammaglobulinemic patients have a complete deficiency of IgG, A and M, but imputing IgA to the facilitation of a translocation is not sufficient. Nearly one person in 600 has a complete IgA deficiency, so other mechanisms likely to promote this infection must be discussed. It should be noted, for example, that in addition to the interruption of treatment (see above), the quality of the immunoglobulin substitution was probably insufficient, with a residual level of 5.3 g per liter.
For the rest, speculations about the role of certain Helicobacter in the occurrence of inflammatory disease of the digestive tract are interesting but perhaps a little off topic unless we focus more specifically on IBD like patients with humoral immune deficiency.
Author Response
Line 66, the authors report treatment with intravenous immunoglobulin every 3 months instead of every 3 weeks.
We agree that the intravenous immunoglobulin treatment should have been every 3 weeks, but according to the patient the frequency was every 3 months.
Some English corrections are required, for example on lines 68 and 74.
We have, to the best of our ability, made corrections as enquired.
Please note a small typographical error on line 160.
We have, to the best of our ability, made corrections as enquired.
With regard to the clinical description, we would like to know the nature and duration of the interruption of immunoglobulin replacement therapy between the patient's leaving the United States and arriving in Denmark, as well as the chronology of the interruption prior to the occurrence of the infection.
The estimated time of the interruption has been added and was primarily due to the time it took the patient to settle in, get a general practioner appointed and through the GP beeing referred to our department.
It would be helpful to clarify the nature of his work on the pig farm because pigs are known to be carriers of certain strains of Helicobacter.
The patient had direct dayli contact with pigs, providing feed and mucking out fecalia. He did report wearing disposible full body suits, gloves and doing careful hand hygiene after work. We believe that this patient was indeed infected via a feco-oral route from the pigs he worked with.
About Figure 1, the CRP and IgG assay curves should be more specifically captioned. In this figure or in the text, the duration of antibiotic treatment should be more clearly indicated.
The legends to the figure was missing and has now been added. The duration of most of the treatments is included in the text, but for graphical reasons it was not feasible to include them directly in the figure, allthough we agree that it would ideally have been usefull.
In the discussion, it seems important to broaden the reflections on IgA deficiency. It is true that agammaglobulinemic patients have a complete deficiency of IgG, A and M, but imputing IgA to the facilitation of a translocation is not sufficient. Nearly one person in 600 has a complete IgA deficiency, so other mechanisms likely to promote this infection must be discussed. It should be noted, for example, that in addition to the interruption of treatment (see above), the quality of the immunoglobulin substitution was probably insufficient, with a residual level of 5.3 g per liter.
This is a good point. We agree and have now broadened the discussion on this topic.
For the rest, speculations about the role of certain Helicobacter in the occurrence of inflammatory disease of the digestive tract are interesting but perhaps a little off topic unless we focus more specifically on IBD like patients with humoral immune deficiency.
A valid point. We have chosen to keep the paragraph, but leave it short.